# Patient-Reported Experience Measures for Colonoscopy: A Systematic Review and Meta-Ethnography

**DOI:** 10.3390/diagnostics12020242

**Published:** 2022-01-19

**Authors:** Annica Rosvall, Magdalena Annersten Gershater, Christine Kumlien, Ervin Toth, Malin Axelsson

**Affiliations:** 1Department of Care Science, Faculty of Health and Society, Malmö University, 214 28 Malmö, Sweden; magdalena.gershater@mau.se (M.A.G.); christine.kumlien@mau.se (C.K.); malin.axelsson@mau.se (M.A.); 2Department of Cardio-Thoracic and Vascular Surgery, Skåne University Hospital, 205 02 Malmö, Sweden; 3Department of Gastroenterology, Skåne University Hospital, Lund University, 205 02 Malmö, Sweden; ervin.toth@med.lu.se

**Keywords:** colonoscopy, endoscopy, meta-ethnography, item–concept mapping, patient experience, patient-reported experience measures, quality measurements, review, triangulation

## Abstract

Patient experience is defined as a major quality indicator that should be routinely measured during and after a colonoscopy, according to current ESGE guidelines. There is no standard approach measuring patient experience after the procedure and the comparative performance of the different colonoscopy-specific patient-reported experience measures (PREMs) is unclear. Therefore, the aim was to develop a conceptual model describing how patients experience a colonoscopy, and to compare the model against colonoscopy-specific PREMs. A systematic search for qualitative research published up to December 2021 in PubMed, Cochrane, CINAHL, and PsycINFO was conducted. After screening and quality assessment, data from 13 studies were synthesised using meta-ethnography. Similarities and differences between the model and colonoscopy-specific PREMs were identified. A model consisting of five concepts describes how patients experience undergoing a colonoscopy: health motivation, discomfort, information, a caring relationship, and understanding. These concepts were compared with existing PREMs and the result shows that there is agreement between the model and existing PREMs for colonoscopy in some parts, while partial agreement or no agreement is present in others. These findings suggest that new PREMs for colonoscopy should be developed, since none of the existing colonoscopy-specific PREMs fully cover patients’ experiences.

## 1. Introduction

Colonoscopy is considered a standard procedure for patients in need of diagnosis, treatment, surveillance, and/or colorectal cancer (CRC) screening [1,2]. The number of colonoscopies performed is mounting due to extended life expectancy together with increased incidence of CRC [3,4]. Even though patients may experience a colonoscopy as unpleasant, embarrassing, uncomfortable, and even painful [5], the acceptance of the procedure is crucial [6]. Seven quality domains have been identified in the current guidelines for lower gastrointestinal endoscopy from the European Society of Gastrointestinal Endoscopy (ESGE); patient experiences being one of them [6]. Patients’ experiences can be defined as ‘to which extent patients are receiving care that is respectful of and responsive to individual patient preferences, needs and values’ [7]. Clearly, the patients’ needs should be reflected in the care they receive and, moreover, patients should be seen as partners whose feedback can enhance the clinical performance [8] and safety [9]. They are, hence, the experts of their own experiences [10] and their perspective is an indicator of care quality [11]. Therefore, it is recommended that patient experiences should be measured routinely due to the importance of acceptance of the colonoscopy and the patient’s willingness to repeat the procedure. However, there is a need for patient-reported experience measures (PREMs) regarding patients’ experiences before, during, and after a colonoscopy procedure. Such an instrument can be used for continuous evaluation of quality improvement projects regarding colonoscopy or for research in the field. In addition, the guidelines recommend that the experiences should be reported by the patients themselves by answering questions that are relevant to them [6].

Brown and colleagues identified endoscopy-specific instruments which were developed and used in studies to evaluate how the patients experienced different endoscopic procedures [12]. In addition, a recent review identified instruments which aimed to measure patient-reported satisfaction and experiences regarding CRC screening, including tests, and relevant procedures such as colonoscopy [13]. However, questions have been raised whether existing instruments succeed in measuring how patients experience undergoing a colonoscopy, since descriptions of patient involvement during instrument development are missing [12,14] and validation of them is sparse [13].

Several reviews have described patient experiences concerning barriers and facilitators surrounding colonoscopy and participation in CRC screening [15,16,17,18,19,20]. However, the results either derive from both quantitative and qualitative data related to patients in a CRC screening context [15,16,18,19] or present findings from a population consisting of both healthcare providers and patients [17]. Accordingly, a review focusing solely on qualitative data and including both patients’ experiences of the CRC screening context and patients with other indications for the colonoscopy, such as diagnosis, treatment, and/or surveillance, is missing.

Patient experiences are fundamental indicators of healthcare quality [11] and those experiences are best measured by empirical evidence from the patient perspective [8,21]. Therefore, a synthesis of qualitative research can contribute to compiled knowledge of how adult patients experience undergoing a colonoscopy procedure. In addition, existing instruments aiming to measure patient-reported experiences in connection with colonoscopy have been criticised due to the lack of patient input during instrument development [12,14] and absent validation [13]. For that reason, a comparison between empirical evidence from a qualitative synthesis and existing instruments is justified.

The aim of this study was to develop a conceptual model by reviewing studies with a qualitative design which explore and describe how adult patients experience a colonoscopy procedure, and to triangulate the conceptual model with existing colonoscopy-specific PREMs.

## 2. Materials and Methods

Meta-ethnography was chosen to synthesise the qualitative studies in the systematic review [22]. A review protocol was registered in PROSPERO on 23 July 2019 and was last updated on 10 March 2021 (CRD42019122422). The systematic review was supported by the PRISMA 2020 checklist [23]. Data that were explicitly linked to how adult patients experienced undergoing a colonoscopy were extracted and synthesised in line with Britten and colleagues [24] and reported according to the eMERGe reporting guidance [25]. Findings from the meta-ethnography were compared, using triangulation as a method, [26] against items in existing colonoscopy-specific PREMs.

### 2.1. Search Strategy and Selection Process

The research question was phrased as What experiences do adult patients have of undergoing a colonoscopy? and broken down using the PEO method [27] (Table 1).

For inclusion, the studies had to have a qualitative research design, be written in English, have an abstract, and be published in a peer-reviewed journal. Additionally, the population had to be adult patients aged ≥18 years who had undergone a colonoscopy. Appropriate search terms were identified in collaboration with an experienced librarian, and this was followed by a systematic search, which was performed, on 11 September 2020, in four databases, PubMed, Cochrane, CINAHL, and PsycINFO, with an updated search on 11 December 2021. Searches were done using subject headings, thesaurus or MeSH together with the Boolean operators OR and AND. The first search resulted in 6256 studies and the updated search in 715 studies, i.e., a total of 6971 (Figure 1).

Duplicates were removed in EndNote [28], resulting in 4867 studies identified for screening. These studies were imported into Rayyan, which is a web application that enables reviewing of abstracts and titles in a blinded mode [29]. Accordingly, the first (AR) and last author (MA) each performed an individual title/abstract screening of the studies, in Rayyan with activated blind mode, resulting in an identification of 23 studies. These studies were assessed for eligibility as specified by a protocol matching the aim of the inclusion criteria. After individual reading of the 23 studies in full text, a consensus was reached for 21 studies and MAG, the second author, was consulted regarding two studies, as a third reviewer. The eligibility assessment resulted in the inclusion of thirteen full-text studies, and these were assessed for quality according to a study-specific protocol [30,31]. Each study was graded in relation to nine assessment criteria which aimed to evaluate high, moderate or low scientific quality [31]. The grading was independently performed by AR and MA and followed by a discussion to secure consensus. Eight of the studies were given a high score in the grading process and therefore considered of high scientific quality, whereas the number of points attributed to five of the studies labelled them as being of moderate quality. Due to a conflict of interest regarding the quality assessment of one study [32], two external reviewers were consulted. The quality assessment resulted in thirteen studies of moderate and high quality and these were all included in the meta-synthesis (Table 2). Detailed study characteristics and key contextual information are available in Appendix B (Table A1).

### 2.2. Data Extraction and Analysis

All thirteen studies were read multiple times by two of the authors (AR and MA), who individually appraised data that responded to the study aim, and this was followed by a consensus discussion between the two of them. Relevant data were extracted and included into the meta-synthesis from specific sections consisting of both quotes from the primary studies and reported findings.

The process of data analysis was iterative, and all authors contributed to the synthesis during recurrent consensus discussions. Initially, workshops (AR and MA) were held, to identify metaphors in the extracted data and to determine how the thirteen studies were related to each other by using a mind-map technique. The workshops resulted, tentatively, in twelve concepts and these concepts were presented to all authors, followed by a consensus discussion. The concepts were then condensed by AR and MA and again discussed among all authors until agreement was reached, which resulted in five main concepts. Data that were associated with a specific concept were organised in a grid. In addition, data which presented patients’ experiences from either before, during, or after the colonoscopy procedure were marked in the grid in accordance with each specific time. Key texts related to each concept were highlighted in the grid and further analysed, one by one, focusing on similar and contrasting aspects. These findings were discussed with all authors and, after general agreement of interpretation, expressed as third-order constructs. The synthesis of the meta-ethnography was expressed in a Line of Argument (Appendix A).

### 2.3. Inclusion of Instruments

Patient experience in connection with colonoscopy seems to include several dimensions [15,16,17,18,19,20] and, therefore, instruments with more than two dimensions are reasonably of interest for this study. For inclusion, the following was required: a published peer-reviewed full-text paper written in English and reporting findings regarding a self-reported multidimensional instrument aiming to measure patient experience in connection with colonoscopy. Exclusion criteria were: case reports, studies aiming to measure patient-reported outcomes, conference papers and studies published before 2001. Several endoscopy-specific patient experience measures (n = 48) were identified in a review with systematic searches from 1980 to November 2013 [12]. Four instruments from this review met the inclusion criteria: the GI Procedure Patient Satisfaction Survey [45], the Global Rating Scale (GRS) [46], the mGHAA-9 [47], and the Patient satisfaction questionnaire [48]. To find further published colonoscopy-specific PREMs, a search was conducted in PubMed between 1 December 2013 and 1 November 2021. The search term for the procedure was colonoscopy, and for the outcomes and measures, the same terms as specified by Brown and colleagues were used [12]. After removing duplicates, 4568 studies were identified. This was followed by a title/abstract screening which was performed by the first author (AR) and resulted in the identification of four new instruments that met the inclusion criteria: the Colonoscopy Questionnaire from NHS’s Bowel Cancer Screening Program (BCSP) [49], the Colonoscopy Satisfaction and Safety Questionnaire (CSSQP) [50], the Gastronet [51], and the Gastrointestinal Endoscopy Satisfaction Questionnaire (GESQ) [52]. All eight instruments were included in the triangulation.

### 2.4. Triangulation

Triangulation was used to explore similarities and differences between the conceptual model from the meta-ethnography and items in existing instruments aiming to capture colonoscopy-specific PREMs [26]. This was done by creating a triangulation protocol where the main concepts in the established conceptual model formed the columns and the instruments formed the rows. Two reviewers (AR and MA) individually mapped how well the colonoscopy-specific PREMs from the included instruments’ items corresponded with the main concepts, by giving agreement scores. No interpretation was made and solely data explicitly stated in the concepts and the items were mapped against each other. When the overlap was complete it was awarded a sufficient score (+). If the mapping showed partial overlapping it was awarded an insufficient score (+/−) and when no overlap was present the mapping resulted in a no agreement score (−).

## 3. Results

### 3.1. Meta-Ethnography

Thirteen studies involving 245 adult participants (10–29 participants per study) were included in the meta-ethnography. The participants’ age ranged from 17 to 85 years; 42% of them were male (n = 103), 32% were female (n = 79) and the age of the remaining 26% (n = 63) was not reported. The studies included participants who were undergoing a screening colonoscopy [34,35,41,42,44], or where the participants’ indication for the colonoscopy was either colonic inflammatory bowel disease or suspected colonic neoplasia [33,36,37]. Two studies included patients within colorectal cancer screening as well as patients with other clinical indications [32,40], one study explicitly stated that the participants had no suspected cancer [43], and two studies had no reporting indication [38,39]. Seven studies clearly stated that some participants had undergone more than one colonoscopy [33,34,37,38,39,41,44]. The data were collected after the colonoscopy procedure, using either individual interviews [32,33,35,36,37,38,39,40,42,43] or focus group interviews [34,41]. One study used both individual and focus group interviews [44]. All but two [40,44] were single-centre studies and conducted in Australia (n = 2), Canada (n = 1), Denmark (n = 3), Sweden (n = 2), the United Kingdom (n = 3), and the United States (n = 2). Three studies compared the patients’ experiences of colonoscopy with other alternative examinations of the colon [33,42,43].

The meta-ethnography resulted in a conceptual model including five main concepts, *Health motivation*, *Discomfort*, *Information*, *A caring relationship,* and *Understanding* (Figure 2) and all included studies contributed content to the synthesis (Appendix A).

#### 3.1.1. Health Motivation

The concept *Health motivation* relates to aspects that motivated the participants to go through with the colonoscopy procedure and this concept was presented in ten of the thirteen included studies [32,34,35,36,38,39,40,41,42,43]. Some patients were terrified of having cancer [32,34] and feared a potential diagnosis and the results of the colonoscopy [36,38,40], while others were unworried and accepted the colonoscopy as an indisputable necessity so that they could get relieved of the fear of having cancer [35] or maintain their health [39]. The latter view was related to an understanding of the colonoscopy as a trustworthy procedure if some tissue [42] or anomalies had to be removed [39]. Patients with symptoms felt that they could not refuse to undergo the colonoscopy, because they needed to find out what was causing them difficulties [36,40] and they urgently desired an understanding of what they had to deal with [32,39]. Some expressed hopeful thoughts about finding a potential cancer diagnosis in time for treatment, because they were not ready to leave earthly life and believed that the colonoscopy increased their chances of survival [32]. In addition, being offered a colonoscopy when in need of it gave rise to feelings of gratitude [40]. Regardless of what reasons motivated the patients to undergo the colonoscopy, they had a desire to be healthy [32,36,38,40,41,42] and to determine their current state of bowel health [35,39,40,43].

#### 3.1.2. Discomfort

All thirteen studies presented findings regarding different aspects of *Discomfort*, either before, during, or after the procedure [32,33,34,35,36,37,38,39,40,41,42,43,44].

*Before*. The period prior to the colonoscopy was challenging, especially for patients with pre-existing conditions since they worried that the procedure could exacerbate those conditions [34,41]. In particular, patients with diabetes experienced the bowel preparation as difficult due to their diet restrictions and fluctuations in their plasma glucose levels [34,41]. Other patients, too, were concerned about the aspect of dietary restrictions, which were experienced as overwhelming due to the planning and preparation of meals that often deviated from their usual routine [40]. Overall, the bowel preparation was experienced as unpleasant [32,35,36,37,38,44] and patients felt nervous when they were facing the intake of the large fluid volume [34]. During the bowel preparation, they could experience nausea and abdominal discomfort [32,40,41,44]. However, they tolerated the preparation due to a desire to do a good job [32,38,44] and their willingness to go through with the colonoscopy since they felt that the procedure had to be done [36]. There were however exceptions, some patients did not experience discomfort during the bowel preparation [44]. The patients also made logistical plans prior to the colonoscopy, as they had to take time away from other duties [32,42,44] and were in need of transportation to and from the hospital [32,36,41,42].

*During*. When the colonoscopy was performed, the patients felt that they were exposed [32,40] and in an awkward [42] situation and they were embarrassed by the procedure’s sensitive nature [37,40]. Embarrassment could be experienced if the physician was of the opposite sex [37] or in connection to the actual penetration of anus [37,40,42]. However, patients felt that wearing dignity shorts eased the feelings of embarrassment [32,37]. The procedure itself was experienced as unpleasant [35] but the patients’ experiences of discomfort and/or pain during the procedure differed [32,44]. They felt a mixture of pain and discomfort [33] which was described as sensory experiences ranging from cramp to pain [37]. Some experienced discomfort and pain continuously [42,43], while others felt it occasionally during the procedure [43]. Certain occasions during the colonoscopy, such as insufflation and intubation of the colon, were described as uncomfortable and painful [32,33,37,43,44]. The overall experience of undergoing a colonoscopy was characterised as multidimensional because of the different sensations it entailed, such as seeing, hearing, feeling, and smelling the procedure [32]. By some patients, sedation was used as a main strategy to tolerate discomfort during the procedure [33,35,43,44]. However, the effect of the sedations varied, and some patients were unsure if they had been sedated or not [37]. Due to health status, not all patients were ideally suited for sedation [43] and some purposely avoided it since they wanted to drive afterwards [42]. In addition, healthcare professionals could, through their guidance, support the patients in controlling the discomfort and pain by, for instance, encouraging calm, relaxing breathing [43], change of position, or a temporary pause [44].

*After*. After the colonoscopy, the patients felt exhausted and they described themselves as hungry but tired [32] with temporary headache [44]. Some did not experience any differences in their abdominal health compared to before the colonoscopy, but some felt that their stomach behaved differently afterwards [32]. These differences were described as bloating, soreness, and a change of bowel habits [32,44].

#### 3.1.3. Information

Experiences of *Information* were reported in eleven studies and this concept refers to the time before, during, and after colonoscopy [32,33,35,36,37,38,39,40,42,43,44].

*Before*. Not all patients felt that the given information prepared them for what would happen, which led them to seek their own information [32,37,38,40]. Family and friends were experienced as a source of information and some of the patients talked naturally to them about the colonoscopy [32,38,39]. Other patients felt strongly that the procedure was surrounded with stigma and constituted an inappropriate topic of conversation [36]. Furthermore, the written information prior to the colonoscopy, about what the patients could expect after intake of bowel preparation, was experienced as limited [37,38,40]. The patients struggled to follow the preparation instructions, and some appealed for verbal confirmation that they had understood them properly [32,44].

*During*. The possibility for immediate sharing of information, either visually via the monitor or verbally, during the colonoscopy, was often perceived as fascinating and positive by the patients [32,33,35,43,44], since it made them feel involved [32,35] and gave them a preliminary evaluation of the result [40]. Nevertheless, some of them thought it was either too explicit or boring to watch the screen [43,44]. Moreover, if the patients had sedation, they could have difficulties fully comprehending the meaning of the images on the monitor or remembering what had been said to them during the procedure [33,43].

*After*. After the colonoscopy, the patients wanted to know the result and felt relieved when getting it [32,35,36,40]. The patients experienced reassurance if a negative result confirmed that no serious illness was present [36,39], and, in contrast to this, frustration if they received a negative result which did not reveal the cause of their symptoms [36] and when there was a lack of a solution to their problems [32,39]. When sedated patients were given information about the test outcome during recovery, this was perceived as positive [38,43]. If the patients, for some reason, were discharged without being verbally informed about the results, they experienced anxiety regarding the outcome and how they would be informed about it afterwards [32,37,40].

#### 3.1.4. A Caring Relationship

Eleven of the included studies described findings concerning *A caring relationship* between healthcare professionals and patients [32,34,35,36,37,38,39,40,42,43,44] during the colonoscopy procedure, but no experiences before or after were reported.

*During*. How the healthcare professionals behaved towards the patients made a great difference to how the patients experienced the colonoscopy [32,35,36,38,39,40,42,43]. However, although a friendly behaviour was a key factor for a positive experience, professionalism was also described as important, giving the patients faith in and making them trust [32,34,35,36,37,43,44] the healthcare professionals’ competence. Thanks to faith in the healthcare professionals, the patients surrendered to the doctors’ control [36]. Although this shift in power was voluntary, the opportunity for the patients to still be in control, and allowed to stop the procedure if needed, was highlighted as valuable [37]. When a caring relationship was created between the patient and the healthcare professionals it was emphasised as an important factor for a beneficial experience [32,36,40]. Examples of relationship-building interactions between the patient and the healthcare professionals were that the patients felt that they were being heard [35,36], and that the healthcare professionals were supportive towards them [39,43] and attentive to the patients’ needs [32,35,40]. Patients experienced attenuated tensions if a positive atmosphere was created with small talk and humour [38,40]. The building of a relationship between the healthcare professionals and the patient made the patients feel respected [32] and safe [40], reducing anxiety [36] and embarrassment [43]. If a respectful interaction was lacking, patients experienced less individual care [42] as well as feelings of insecurity [40].

#### 3.1.5. Understanding

The concept *Understanding* consists of data from nine studies [32,34,36,37,38,39,40,41,44] describing pre- and post-experiences of going through a colonoscopy. The result of this concept thus contains experiences before and after, but none during the procedure.

*Before*. Patients relied on the healthcare professionals being competent [40]. However, prior to the colonoscopy, patients experienced a variety of worries [38,44], such as fear of suffering from potential complications, related to either perforation [34] or difficulties breathing properly [41], or even death [36]. In advance, patients often experienced uncertainty about what to expect [32]. Furthermore, if the patients had experiences that disharmonised with their expectations regarding how sedated they should be during the colonoscopy, this created a conflict which troubled them in a negative way and influenced how they experienced the procedure afterwards [39]. Moreover, patients with former experiences of undergoing a colonoscopy knew what to expect and were less concerned than those who were about to undergo the procedure for the first time [39,44]. However, previous negative experiences of a colonoscopy served as a barrier to going through with the procedure [41] and patients with these experiences were more anxious [37]. Patients described a colonoscopy as being a procedure surrounded by negative attitudes, such as awkwardness, and when they appraised the colonoscopy in advance, both personal and social attitudes influenced their understanding of it [36].

*After*. The beliefs about colonoscopy before the procedure do not necessarily correspond to the actual experience of it [36,44]. When the patients had undergone the colonoscopy, they looked back in time and reflected on the whole experience and often concluded that their worries beforehand had been unfounded since their personal experiences of undergoing the procedure had been fairly unthreatening [32,36,38,40,44]. In contrast, if the bowel preparation had been tougher than expected [44] or the colonoscopy had been more painful than anticipated, this affected their attitude to the procedure and their willingness to repeat a colonoscopy in the future in a negative way [38,39]. Nevertheless, to undergo a colonoscopy is preparatory for future procedures [40,44].

#### 3.1.6. Line of Argument

Adult patients’ experience of undergoing a colonoscopy is illustrated by means of a conceptual model, which can be defined according to a Line of Argument:

Experienced good health is a desirable state of being. A colonoscopy is an inconvenient procedure for the patients which can be alleviated by the support of healthcare professionals. Beneficial patient experiences are created by sharing of information. Mutual respect and trust are the foundation for a caring relationship between the patient and healthcare professionals. The patients’ understanding prior to the colonoscopy is re-evaluated by them after the experienced procedure.

### 3.2. Triangulation

The triangulation between the conceptual model and existing instruments, presented in Table 3, showed that none of the existing instruments measure the variety of experiences of undergoing a colonoscopy reflected in the conceptual model. Of the eight instruments, the mGHAA-9 [47] and the Colonoscopy Questionnaire BCSP-NHS [49] covered four of the concepts, although not with regard to the whole colonoscopy process. The concept *Discomfort* was mostly covered during the colonoscopy, and in all the instruments, the concept *A caring relationship* was, just as in the conceptual model, covered during the procedure. *Information* was mainly covered before and after the colonoscopy. Instruments consisting of items that covered the concept *Understanding* were sparse, especially after the colonoscopy, and none of the instruments consisted of items reflecting the overall concept *Health motivation.*

## 4. Discussion

The first part of the current study, the meta-ethnography, showed that undergoing a colonoscopy is surrounded by several varieties of experiences related to health motivation, discomfort, information, a caring relationship, and understanding. The second part, the triangulation, showed that none of the available existing colonoscopy-specific instruments to date cover all these experiences.

The conceptual model in the meta-ethnography clearly demonstrates that there are distinct experiences, although they are interrelated, as illuminated in the Line of Argument and depicted in Figure 2, which argues for all these aspects being included in colonoscopy-specific PREMs. *Health motivation* drives the patients to go through with the colonoscopy, which is an important finding as this could be used to prepare and motivate patients to overcome barriers. Individuals’ lack of knowledge of current colonoscopy guidelines [15,19] and their poor understanding of the advantages of a screening colonoscopy [16,20] have previously been reported as barriers to undergoing the procedure. Deficient information about screening colonoscopy might cause misconceptions affecting the decision to participate [18]. In contrast, awareness about colorectal cancer was highlighted as positive and can facilitate participation in screening [15,16,17]. A fundamental desire to experience good bowel health was emphasised in the meta-ethnography and, interestingly, none of the instruments in the triangulation included items related to health motivation.

The conceptual model reveals that *Discomfort* was experienced throughout the whole colonoscopy process. This is not captured in existing instruments, which primarily focus on discomfort during colonoscopy [45,46,47,49,50,51,52], although some include items about discomfort prior to the procedure [45] and afterwards [45,49]. Furthermore, little is known of how the patients experience the time after the procedure with regard to discomfort. One-third of patients who have undergone a colonoscopy experience minor adverse events in the first 1–2 weeks [53]. Abdominal discomfort and bloating are most common [53] and this is in line with findings from two of the studies included in the meta-ethnography [32,44]. However, the conceptual model adds knowledge regarding how patients experienced a need to recover both emotionally and physically after the procedure. This indicates that new colonoscopy-specific PREMs need to include items that reflect both physical and emotional aspects of discomfort.

Patients’ experiences regarding *Information* are well covered in the conceptual model—before, during and after the colonoscopy. All instruments contained items about information before and after, while, surprisingly, the time during the colonoscopy was less covered. Some studies have concluded that non-pharmacological interventions, such as auditory, verbal, and/or visual information, can work as distractions to reduce anxiety during the procedure [54] and enhance patient experience [55]. Regardless, healthcare professionals should consider individual preferences, since patients experience a variety of emotions during the procedure [18] and may perceive and handle information differently [56]. However, patients who receive sedation might not comprehend specific events, or relevant information given to them, during the colonoscopy [57]. Nevertheless, and not least regarding unsedated patients, the arguments for including questions on how they experienced the given information during the procedure are difficult to ignore for an instrument claiming to measure colonoscopy-specific patient experiences.

The conceptual model lacks findings regarding *A caring relationship* both before and after the procedure, which might be considered logical since the time before and after is in many ways characterised by self-care. Whether the patients would prefer a caring relationship or not before and after their colonoscopy is unclear and needs further exploration. Nevertheless, the conceptual model describes ingredients for a caring relationship and many of those are to be found in existing instruments. Thus, existing instruments include items focusing on the healthcare professional’s respectful behaviour towards the patients [46,47,48,49,50,51,52], on whether the patients felt that they trusted the healthcare professionals [47,51,52] and on whether the healthcare professionals had listened to the patients’ needs [45,46,49]. To create a caring relationship, trust needs to be achieved through dialogue and sharing of information between the patient and the healthcare professional [58]. Patient participation is a concept that includes a caring relationship in which learning and reciprocity occur [58]. Several factors which may facilitate patient participation throughout the endoscopy pathway were identified during interviews with patients who had undergone an endoscopic procedure such as colonoscopy [59]. When, for instance, patients felt acknowledged as individuals with their own expectations and fears, by the healthcare professionals, they experienced involvement and this was reported as an example of patient participation in a clinical context [59]. A caring relationship between patients and healthcare professionals being the essence of patient participation [58], future colonoscopy-specific instruments should, this review suggests, further develop items aiming to measure this.

Not many existing instruments had items reflecting the concept *Understanding*, especially not with regard to the time after the colonoscopy. The concept describes both how the patients anticipated that they would experience the impending colonoscopy and their subsequent careful reflections about how they in fact experienced undergoing the procedure. The actual experience of undergoing a colonoscopy is often undemanding compared to the negative expectations many patients experience prior to the procedure [5]. The patients’ understanding of the colonoscopy paves the way for a positive experience and that is why, for instance, items regarding former colonoscopy experiences should be included in a colonoscopy-specific PREM.

A strength with the current study is that the method is meticulously accounted for and that the stages in the method have been systematically performed, enabling replication. In addition, triangulation has been used to ascertain whether the empirical evidence complies with existing colonoscopy-specific PREMs and these findings can serve as a foundation for the development of new measures of patient-reported experiences of colonoscopy. Another strength is that this study is a synthesis of the experiences of a large group of 245 patients from different settings/countries, who, in previous qualitative research, have shared their experiences. This argues for these findings covering experiences of importance that can be used in the development of a new instrument. The aim was to develop a conceptual model and, since an understanding was wanted, meta-ethnography was chosen, this being a method that can advantageously be used when the aim is to develop a conceptual understanding of a phenomenon [22]. The conceptual model presents an overview of patients’ experiences related to colonoscopy. As discussed, additional research, on especially the concepts of a caring relationship and of understanding, may contribute to an even clearer view of the whole process of how patients experience undergoing a colonoscopy. Nevertheless, this review comprises patients from a screening context as well as patients with clinical indications, synthesising qualitative data where the patients themselves have described their experiences after undergoing a colonoscopy. The findings could therefore constitute a solid scaffolding for and support the development of new measures for colonoscopy-specific patient-reported experiences. A potential limitation of our paper is that the included studies used different qualitative methodologies, which may have affected the synthesising of data. Another limitation may be the inclusion criteria for the existing instruments, since they may have led to missed colonoscopy-specific PREMs due to the fact that only instruments with more than two dimensions were included in this study. Despite patients evidently experiencing more than two dimensions when they undergo a colonoscopy, there are several instruments measuring only one or two dimensions, which may succeed in capturing parts of colonoscopy-specific experiences, but not the whole process [60,61,62,63,64].

## 5. Conclusions

The current study strongly suggests that a new instrument reflecting a more comprehensive variety of colonoscopy-specific PREMs needs to be developed and that patients with experiences of undergoing colonoscopy procedures should be involved in the creation of it.

## Figures and Tables

**Figure 1 diagnostics-12-00242-f001:**
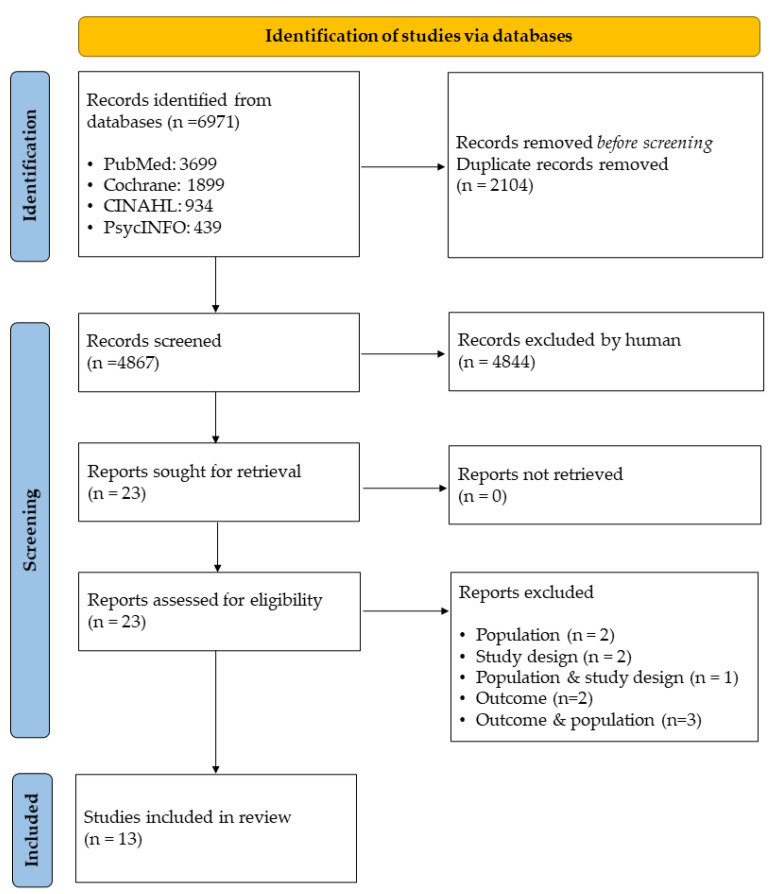
PRISMA Flow diagram of included studies (n = 13).

**Figure 2 diagnostics-12-00242-f002:**
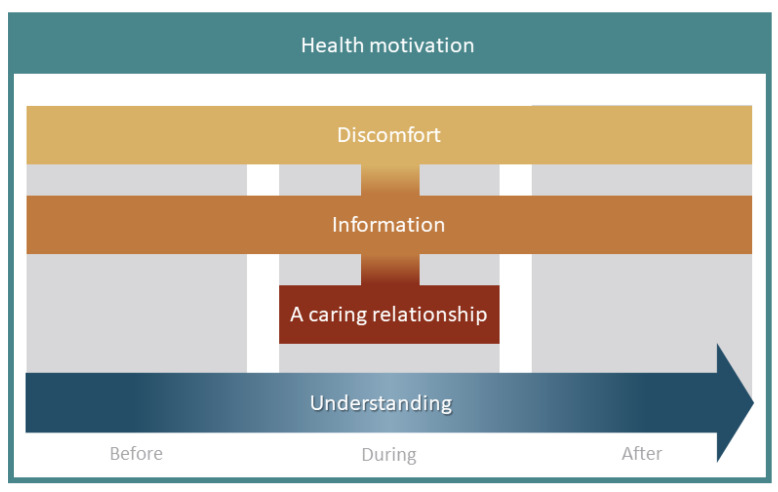
Conceptual model with five main concepts from the meta-ethnography findings.

**Table 1 diagnostics-12-00242-t001:** Research question broken down using the PEO (population, exposure, outcome) method.

Population	Exposure	Outcome
Adult patients who have undergone a colonoscopy	A colonoscopy	The patients’ experiences of the colonoscopy

**Table 2 diagnostics-12-00242-t002:** Characteristics of included studies (n = 13).

Study	Sample	Data Collection	Data Analysis	Setting	Quality ^1^
Hafeez et al., 2012 [33]	18 patients	Individual interviews	Thematic analysis	United Kingdom	Moderate
Kimura et al., 2014 [34]	13 patients	Focus group interviews	Thematic analysis	United States	Moderate
Kirkegaard et al., 2019 [35]	22 patients	Individual interviews	Thematic analysis	Denmark	High
Mikocka-Walus et al., 2012 [36]	13 patients	Individual interviews	Thematic analysis	Australia	High
Neilson et al., 2020 [37]	10 patients	Individual interviews	Thematic analysis	United Kingdom	Moderate
Restall et al., 2020 [38]	24 patients	Individual interviews	Qualitative interpretive description methodology	Canada	High
Rollbusch et al., 2014 [39]	16 patients	Individual interviews	Thematic analysis	Australia	High
Rosvall et al., 2021 [32]	24 patients	Individual interviews	Thematic analysis	Sweden	High
Shamim et al., 2021 [40]	25 patients	Individual interviews	Inductive content analysis	Denmark	High
Sultan et al., 2017 [41]	23 patients	Focus groups interviews	Inductive grounded approach	United States	High
Thygesen et al., 2019 [42]	10 patients	Individual interviews	Phenomenological- hermeneutical method	Denmark	Moderate
von Wagner et al., 2009 [43]	18 patients	Individual interviews	Thematic analysis	United Kingdom	Moderate
Wangmar et al., 2021 [44]	29 patients	Focus groups interviews Individual interviews	Inductive content analysis	Sweden	High

^1^ High or moderate quality means that the study has fulfilled most of the criteria for scientific quality [31].

**Table 3 diagnostics-12-00242-t003:** Triangulation protocol. Item–concept mapping (+ agreement, +/− partial agreement, − no agreement).

Instruments	Health Motivation n = 10	Discomfort n = 12			Information n = 10			A Caring Relationship n = 10			Understanding n = 8		
	Overall	Before	During	After	Before	During	After	Before	During	After	Before	During	After
CSSQP [50]	−	−	+	−	+	+	+	−	+	−	−	−	−
Colonoscopy Questionnaire BCSP-NHS [49]	−	−	+	+	+	−	+	−	+	−	+	−	−
Gastronet [51]	−	−	+	−	+/−	−	+	−	+	−	−	−	−
GESQ [52]	−	−	+	−	+	−	+	−	+	−	−	−	−
GI Procedure Patient Satisfaction Survey [45]	−	+	+	+	+	−	+	−	+	−	−	−	−
Global Rating Scale (GRS) ^1^ [46]	−	−	+	−	+	+/−	+	−	+	−	−	−	−
mGHAA-9 [47]	−	−	+	−	+	−	+	−	+	−	+	−	+
Patient satisfaction questionnaire [48]	−	−	−	−	+	+	+	−	+	−	+	−	−

^1^ The Joint Advisory Group (JAG) is owner of the GRS.

## Data Availability

All data in this study were obtained from already published material in scientific journals, referenced in the paper, and can be obtained by any individual with access to these journals. Search strings for the meta-ethnography and for the existing instruments are accessible upon request from the corresponding author.

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
