# Peer review of "Patient-Reported Experience Measures for Colonoscopy: A Systematic Review and Meta-Ethnography"

_diagnostics, 2022, doi:10.3390/diagnostics12020242_

Round 1

Reviewer 1 Report

Very detailed review description with meticulous methodology in the final stage before publication, although a concrete PREM for colonoscopy instrument suggestion is lacking.   Nothing to improve in my opinion.

Reviewer 2 Report

This is an overall good study. The idea is valid, the subject important, the published material supports the research, the plan is appropriate, the results support the conclusion and the presentation is excellent. On the other hand the studies included carry small numbers and the groups are highly eterogeneous. For example, IBD patients and colorectal operations patients are probably enough informed and maybe experience more discomfort than the screening patients. 

Reviewer 3 Report

This study suggests a new model measuring how patients experience undergoing a colonoscopy

It consists of five concepts; health motivation, discomfort, information, a caring relationship and understanding. The authors compared the new model with existing colonoscopy-specific PREMs and the similarities and differences between this model and existing PREMs.

The development of new measurement model, and the comparison with existing measurements were appropriate. The new instrument focuses on the patient experience and comprehensively includes existing colonoscopy-specific measurement tools. It seems to be clinically helpful.